# Resolution Enhancement of Spherical Wave-Based Holographic Stereogram with Large Depth Range †

**Zi Wang** [1,2], **Guoqiang Lv** [2,*], **Miao Xu** [1,*] , **Qibin Feng** [1], **Anting Wang** [3] **and Hai Ming** [3]

1    National Engineering Laboratory of Special Display Technology, Special Display and Imaging Technology Innovation Center of Anhui Province, Academy of Opto-electric Technology, Hefei University of Technology, Hefei 230009, China; wangzi@hfut.edu.cn (Z.W.); fengqibin@hfut.edu.cn (Q.F.)

2    Anhui Province Key Laboratory of Measuring Theory and Precision Instrument, School of Instrumentation and Opto-Electronics Engineering, Hefei University of Technology, Hefei 230009, China

3    Department of Optics and Optical Engineering, University of Science and Technology of China, Hefei 230026, China; atwang@ustc.edu.cn (A.W.); minghai@ustc.edu.cn (H.M.)

\*    Correspondence: guoqianglv@hfut.edu.cn (G.L.); xumiao0711@hfut.edu.cn (M.X.)

†    It is an invited paper for the special issue.

**Abstract:** The resolution-priority holographic stereogram uses spherical waves focusing on the central depth plane (CDP) to reconstruct 3D images. The image resolution near the CDP can be easily enhanced by modifying three parameters: the capturing depth, the pixel size of elemental image and the focal length of lens array. However, the depth range may decrease as a result. In this paper, the resolution characteristics were analyzed in a geometrical imaging model, and three corresponding methods were proposed: a numerical method was proposed to find the proper capturing depth; a partial aperture filtering technique was proposed after reducing pixel size; the moving array lenslet technique was introduced after increasing focal length and partial aperture filtering. Each method can enhance resolution within the total depth range. Simulation and optical experiments were performed to verify the proposed methods.

**Keywords:** holographic stereogram; spherical wave; depth range; resolution enhancement; aperture filtering

## 1. Introduction

Holography is one of the most promising three-dimensional (3D) display techniques which can reproduce very realistic 3D images with all depth cues. For electronic holography, the computer-generated holography (CGH) technique is used to calculate hologram patterns from the 3D data of objects [1–3]. Among the many CGH techniques, the holographic stereogram (HS) is an excellent approach for the processing of occlusion culling, gloss reproduction and surface shading, which are important for realistic 3D display. Furthermore, the HS printing technique enables large-scale holographic 3D display [4,5].

The HS is a kind of holographic 3D display based on light field reproduction [6,7]. It is usually spatially segmented into many hologram elements (hogels), and each hogel is the Fourier transform of the corresponding parallax image. Non-hogel-based calculation for HS generation is also possible [8–11]. The parallax images can be obtained by either a camera array or the computer graphics rendering techniques. In reconstruction, the light-rays from all hogels are reproduced to form the light field. The principle of HS is similar to the integral imaging (II) which uses a lens array to reproduce the light field [12–14]. Thus, the HS can be converted from II through fast Fourier transforming (FFT) the elemental images (EI) into hogels [15].

The light-ray sampling and the diffraction at the hologram surface in HS cause image blur far from the hologram plane. Several techniques were reported to improve the image quality of HS for deep scene by adding depth information [16–20]. However, the



acquisition of depth information is not easy. The ray sampling (RS) plane method enhances the resolution by setting the RS plane near the object to sample high resolution projection image [21,22]. The moving array lenslet technique (MALT) was used to enhance the sampling rate of HS without decreasing the angular resolution [23]. The compressive light field could be encoded into a hologram to relieve the spatio-angular resolution trade-off in conventional HS [24]. Recently, we have proposed a new concept of HS which is called the resolution priority HS (RPHS) [25]. The RPHS is obtained by adding a quadratic phase term on the conventional FFT of EIs. Different from conventional HS which uses multiple plane waves to reproduce light field, the RPHS uses spherical waves focusing on the central depth plane (CDP) to reconstruct 3D images. The spherical waves form smaller volume pixels (voxels), thus resulting in enhanced resolution.

In this paper, the resolution of RPHS is analyzed in a geometrical imaging model. Three parameters, capturing depth, pixel size of EI and focal length of the lens array, mainly determine the resolution. It is found that simply modifying the three parameters may enhance the image resolution near the CDP, at a price of decreased depth range. To enhance resolution within the total depth range, three methods are proposed. Firstly, A numerical method is proposed to find the proper capturing depth. Secondly, after reducing pixel size of EI for the resolution enhancement near the CDP, a partial aperture filtering technique is proposed to limit the light ray width for resolution enhancement away from the CDP. Lastly, after increasing focal length of the lens array for the resolution enhancement near the CDP, partial aperture filtering is used to limit the light ray width, which results in dark gaps. The moving array lenslet technique is introduced to fill in the dark gaps.

## 2. The Resolution Characteristics of RPHS

Figure 1 shows the principle of RPHS. Based on the conventional HS which converts each EI to the corresponding hogel through FFT, the RPHS multiplies each transformed hogel with a converged spherical wave phase:

$$H(u,v) = \exp\left[\frac{-jk}{2L}(u^2 + v^2)\right] \cdot A\left(\frac{u}{\lambda f}, \frac{v}{\lambda f}\right), A(f_x, f_y) = FFT[I(x,y)] \tag{1}$$

where *FFT*[] is the fast Fourier transform, $A(f_x, f_y)$ is the frequency spectrum of elemental image $I(x, y)$, $k$ is the wave number, $f$ is the focal length and $L$ is the distance between hologram and the image reference plane (IRP) [25]. Assuming each elemental image has $N \times N$ pixels with pixel size of $\Delta x_1$, and the pixel size of the hogel is $\Delta x_2$, the following equation is derived according to sampling theorem and FFT theory:

$$N\Delta x_1 \Delta x_2 = \lambda f. \tag{2}$$

Generally, $\Delta x_1 = \Delta x_2$, and the size of hogel is equal to the EI. Then, all the transformed hogels can be spliced seamlessly. In RPHS, each hogel emits spherical waves which are focused on the IRP to reconstruct the 3D images. The object points at the IRP are reconstructed perfectly and objects' points around the IRP are reconstructed with good quality. Thus, the IRP is also called the central depth plane (CDP), which means the center of the 3D images. That is to say, the 3D image needs to be located near the CDP to obtain the best reconstruction quality. Thus, $L$ can also be called the capturing depth. At the CDP, the image spot size $S$ is given by lens imaging amplification:

$$S = \frac{L}{f}\Delta x_1. \tag{3}$$

Therefore, by reducing $L$, reducing $\Delta x_1$ or increasing $f$, the resolution at the CDP will be enhanced. However, considering the reconstructions away from the CDP, the situation becomes complicated.

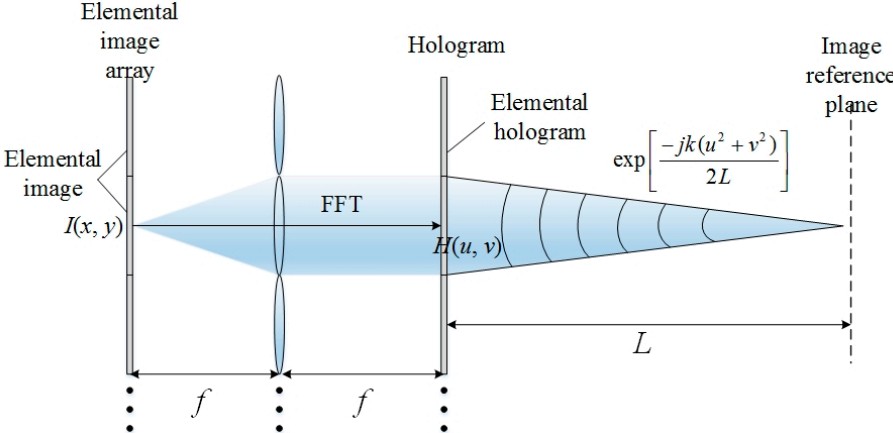

**Figure 1.** The principle of RPHS.

　　Figure 2 shows the image spot size at different reconstruction planes. At the CDP, the focused image spot is formed with the minimum size. At a reconstruction plane away from the CDP, the defocused spot is formed. The image spot size $S$ at reconstruction depth $z$ can be derived easily by dividing the defocused spot into two parts: the green part and the red part. The green part is related with the focused image spot (represented by the blue rectangle) in a similar triangle relation, and the red part is related with the hogel size $p$ in another similar triangle relation. Thus, the image spot size $S$ at reconstruction depth $z$ can be expressed as:

$$S = \frac{L + \Delta z}{f} \Delta x_1 + \frac{|\Delta z|}{L} p, \tag{4}$$

where $\Delta z$ is the defocused depth, and $p$ is the hogel size and is equal to $N\Delta x_2$. The value of $\Delta z$ is negative if $z$ is smaller than $L$, and it is positive if $z$ is greater than $L$. The first term in Equation (4) is indicated by the green rectangle and the second term is indicated by the red rectangle in Figure 2. The first term is related with imaging while the second term is caused by defocusing. Substituting Equation (2) into Equation (4), we can get:

$$S = \frac{L + \Delta z}{f} \Delta x_1 + \frac{|\Delta z| \lambda f}{L \Delta x_1}. \tag{5}$$

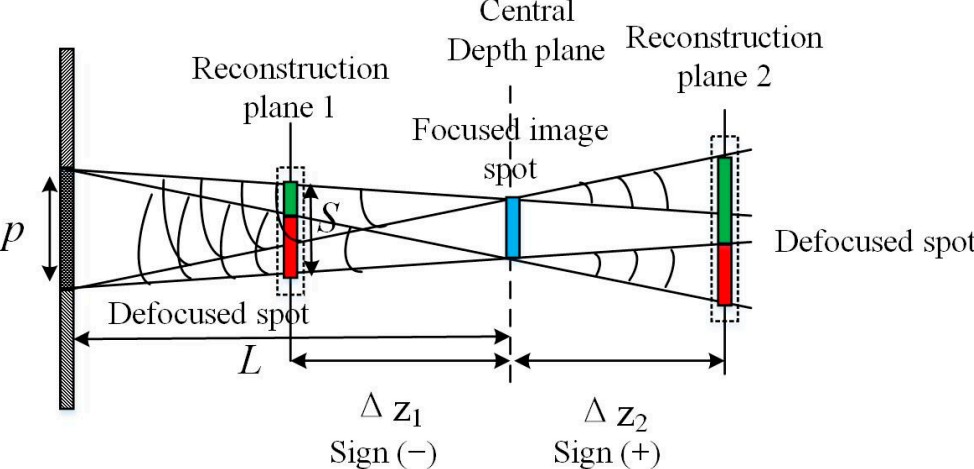

**Figure 2.** The image spot size at different reconstruction planes.

　　Equation (5) shows that at a given defocused depth $\Delta z$, the image spot size is mainly determined by three parameters: CDP depth $L$, focal length $f$ and pixel size $\Delta x_1$. It is interesting that in Equation (5), the values of the first and second terms have opposite changing

directions when the three parameters change. Thus, simply changing the parameters may not work for resolution enhancement. Next, we will show how to enhance resolution by changing the three parameters properly with additional techniques.

### 3. Reducing CDP Depth *L*

Reducing *L* means to capture the 3D object at a closer distance. It is apparent that by reducing *L*, the first term in Equation (5) will decrease while the second term will increase. Thus, for the reconstructed 3D image near the CDP, reducing *L* will always be beneficial to the resolution enhancement. But for the reconstruction away from the CDP, reducing *L* may have a negative influence on the resolution. Reducing *L* is effective only when the reduction of the first term exceeds the increment of the second term. At a small $\Delta z$, it is always satisfied. However, as $\Delta z$ increases, this condition needs to be examined. The limiting case occurs when the reduction of the first term equals the increment of the second term at a given $\Delta z$. In this case, the resolution is enhanced within the total depth $\Delta z$, and an appropriate *L* should be chosen.

Assuming the parameters are $L = 80$ mm, $f = 3$ mm, $\Delta x_1 = \Delta x_2 = 5$ μm, $\lambda = 500$ nm, the relationship between image spot size *S* and defocused depth $\Delta z$ is drawn by the black solid line in Figure 3a. It can be seen that at the CDP, the image spot has the minimum size. As $|\Delta z|$ increases, the image spot size also increases, resulting in a degraded resolution. To enhance resolution, we try to reduce *L* to 30 mm, which is shown by the green dash-dotted line. The resolution at the CDP is greatly improved, but at $|\Delta z| = 20$ mm, the resolution is degraded, indicated by the upward arrow. Thus, the resolution is enhanced at a price of decreased depth range. This case can be avoided by choosing a proper *L*. As shown by the red dashed line with $L = 45$ mm, the resolution at the CDP is improved compared with $L = 80$ mm. In addition, at $|\Delta z| = 20$ mm, its image spot size just equals that of $L = 80$ mm. It means that the resolution is enhanced within the total depth $\Delta z$, without the price of decreased depth range. It is just the limiting case as mentioned above, with an appropriate *L*. The suitable value of *L* can be found by a numerical method. Figure 3b shows the relationship between L and image spot size at $\Delta z = 20$ mm. First, the image spot size with $L = 80$ mm can be obtained, represented by the red point. Then, a horizontal line starting from the red point is generated, and it intersects with the curve to form a green point. The *x*-coordinate of the green point is just the proper value of *L*.

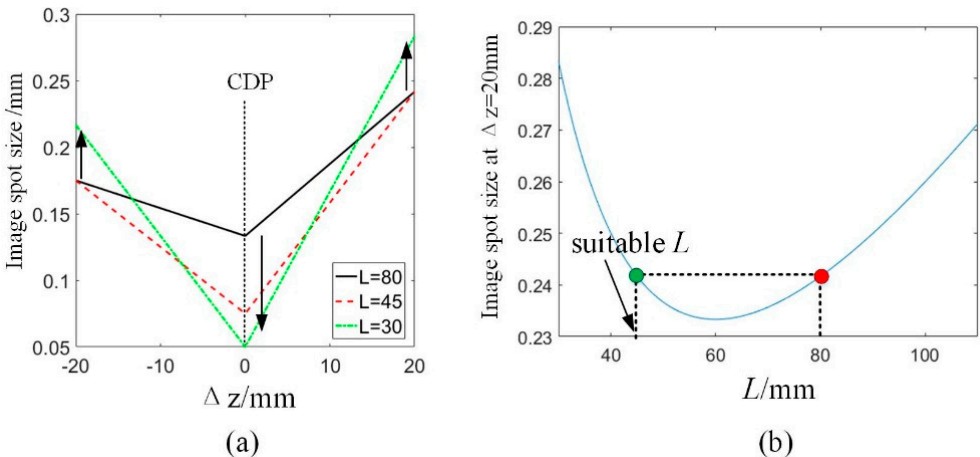

**Figure 3.** (**a**) The relationship between image spot size *S* and defocused depth $\Delta z$ at different capturing depth *L*, with $\Delta x_1 = 5$ μm. (**b**) The relationship between *L* and image spot size at $\Delta z = 20$ mm, with $\Delta x_1 = 5$ μm.

However, the proper value of *L* cannot be found at any situation. Figure 4a shows the case with the same parameters as Figure 3 except for $\Delta x_1 = 3$ μm. It can be seen that reducing *L* will always decrease the resolution away from the CDP, indicated by the upward arrows. On the contrary, Figure 4b shows the case with $\Delta x_1 = 8$ μm. It can be seen that

reducing $L$ will always enhance the resolution within the total depth range, indicated by the downward arrows. These two results can be easily understood in Figure 4c, which shows the relationship between $L$ and image spot size at $\Delta z = 20$ mm. In the case of $\Delta x_1 = 3$ μm, represented by the black solid line, reducing $L$ will cause the increase of the image spot size at the marginal depth, which means reducing $L$ will always cause decreased depth range. Constrastingly, in the case of $\Delta x_1 = 8$ μm, represented by the red dashed line, reducing $L$ will cause the decrease of the image spot size at the marginal depth, which means reducing $L$ is always effective. In both cases, the proper value of $L$ cannot be found using the method in Figure 3b.

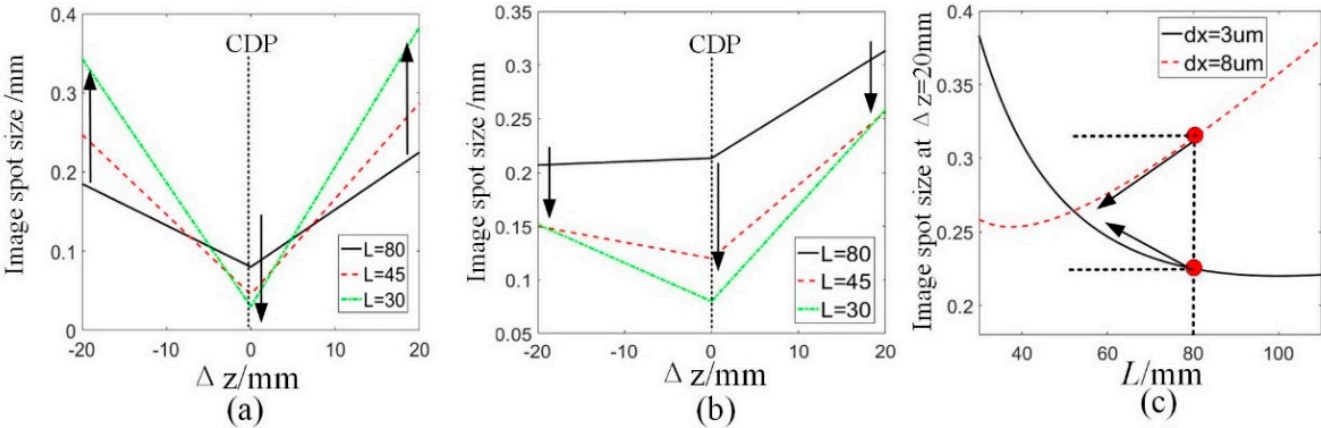

**Figure 4.** The relationship between image spot size $S$ and defocused depth $\Delta z$ at different capturing depth $L$, with (**a**) $\Delta x_1 = 3$ μm and (**b**) $\Delta x_1 = 8$ μm. (**c**) The relationship between $L$ and image spot size at $\Delta z = 20$ mm.

The differences among Figures 3a and 4a,b can also be understood through different weights of the first and second terms in Equation (5). When $\Delta x_1 = 5$ μm and $L = 80$ mm, the two terms have similar weights at large $\Delta z$. By reducing $L$, the reduction of the first term is comparable to the increment of the second term. However, when $\Delta x_1 = 3$ μm and $L = 80$ mm, the second term has greater weight. By reducing $L$, the reduction of the first term is always less than the increment of the second term at large $\Delta z$. That is why reducing $L$ will always decrease the depth range in this case. When $\Delta x_1 = 8$ μm and $L = 80$ mm, the first term has greater weight. By reducing $L$, the reduction of the first term is always more than the increment of the second term within the total depth range. That is why reducing $L$ is always effective in this case. Thus, in different cases, the strategy of reducing $L$ is different.

Next, numerical simulation is performed. Figure 5a shows a bee model and a school badge built in the 3 ds Max modeling software. The camera array consists of $60 \times 60$ virtual pinhole cameras, and the pitch and focal length are set as $p = 0.3$ mm and $f = 3$ mm. The captured EI array contains $3600 \times 3600$ pixels with 5 μm pixel pitch. The CDP is set at the center of the bee model, and the distance between the school badge and CDP is set as $\Delta z = 20$ mm. Figure 5b shows the numerical reconstruction with capturing depth $L = 80$ mm. By reducing $L$ to 45 mm, Figure 5c shows better reconstruction quality at the CDP (see the foreleg of the bee), with comparable reconstruction quality at the marginal depth plane (the school badge). Further reducing $L$ to 30 mm, Figure 5d shows the best reconstruction quality at the CDP. However, the reconstruction quality of the school badge is degraded, which means the depth range is decreased. This simulation verifies that a proper $L$ needs to be chosen to enhance the resolution without decreasing the depth range.

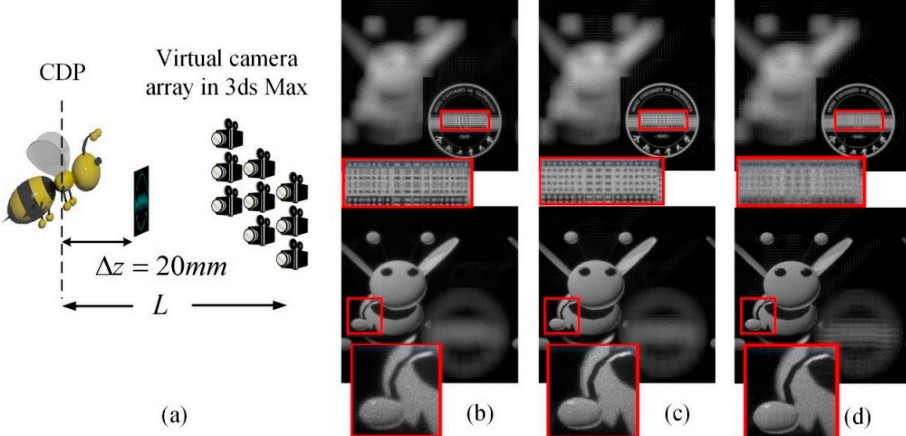

**Figure 5.** (**a**) A bee model and a school badge built in the 3 ds Max modeling software. The numerical reconstruction with capturing depth (**b**) $L$ = 80 mm, (**c**) $L$ = 45 mm, and (**d**) $L$ = 30 mm.

Next, an optical experiment is performed on a binary amplitude-only hologram which is printed on a glass substrate coated with chromium film. The amplitude-only hologram $I(u, v)$ is encoded as:

$$I(u,v) = 2\mathrm{Re}[H(u,v)r^*(u,v)] + C, \tag{6}$$

where $r(u, v)$ is the reference plane wave with incident angle of 3° and $r^*(x, y)$ is the conjugation of $r(u, v)$. $C$ is a constant real value to make $I(u, v)$ non-negative. Then the hologram is binarized by setting the mid-value as the threshold. The hologram contains $3600 \times 3600$ pixels with 5 μm pixel pitch. A solid laser with wavelength of 510 nm was used to illuminate the hologram after being collimated by a lens. Figure 6a–c show the reconstruction results with capturing depth $L$ = 80 mm, 45 mm and 30 mm, respectively. It is easily confirmed that reducing $L$ will always improve the reconstruction quality near the CDP (see the bee model). But to avoid the quality degradation at the marginal depth (see the school badge), a proper $L$ should be chosen.

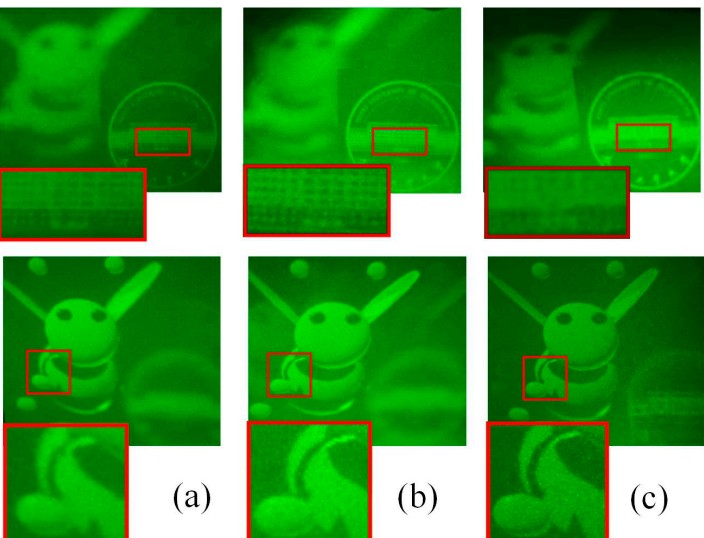

**Figure 6.** The optical reconstruction results with capturing depth (**a**) $L$ = 80 mm, (**b**) $L$ = 45 mm, and (**c**) $L$ = 30 mm.

## 4. Reducing Pixel Size $\Delta x_1$ with Aperture Filtering

According to Equation (5), reducing pixel size $\Delta x_1$ will reduce the value of the first term, or enhance the resolution at the CDP. However, the increase of the second term may decrease the depth range, similar to reducing $L$ in the above analysis. Instead of choosing

a proper value of $\Delta x_1$, we propose to use an aperture filtering technique to enhance the resolution more effectively.

Figure 7a shows a typical case in which the pixel size $\Delta x_1$ of EI equals the pixel size $\Delta x_2$ of hogel. In Figure 7b, $\Delta x_1$ is reduced and according to Equation (2), the hogel size $p = N\Delta x_2$ is increased accordingly. Since the hogel size is larger than EI, the adjacent hogels will have overlap. It can be seen that the focused image spot at the CDP is decreased, making a better reconstruction quality near the CDP. However, the increased hogel size will cause wider light rays, making larger defocused spots away from the CDP. As a result, the depth range is decreased. To address this issue, an aperture filter is introduced to limit the width of the light ray in Figure 7c. The calculated complex amplitude distribution of each hogel is multiplied with a rectangle function Rect($u/p_0$)·Rect($v/p_0$), where $p_0$ is the pitch of the EI. Then each filtered hogel can be spliced seamlessly. The aperture filter reduces the width of light rays, making better reconstruction qualities within the total depth range.

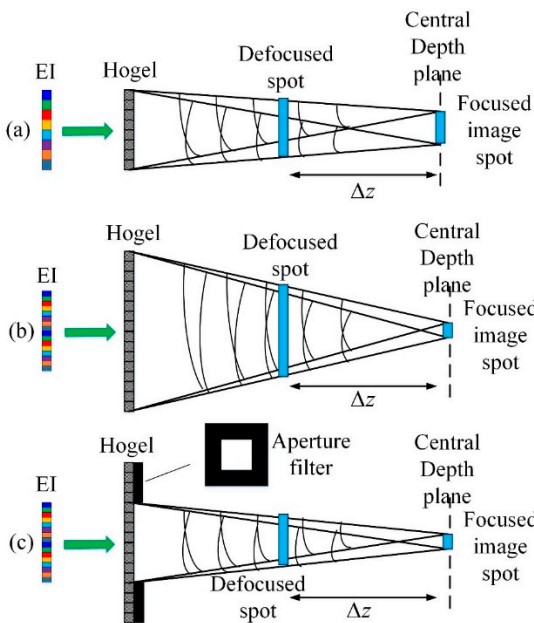

**Figure 7.** (**a**) A typical case in which the pixel size $\Delta x_1$ of EI equals to the pixel size $\Delta x_2$ of hogel. (**b**) Reduced $\Delta x_1$ causes enlarged hogel size. (**c**) The proposed aperture filtering technique.

Figure 8a–c show three groups of numerical simulation results corresponding to the three cases in Figure 7. In Figure 8a, the parameters are set as $p = 0.3$ mm, $f = 3$ mm, $\Delta x_1 = \Delta x_2 = 5$ μm, $L = 80$ mm. The captured EIA consists of $60 \times 60$ EIs, each with $60 \times 60$ pixels. The CDP is set at the center of the bee model, and the distance between the school badge and CDP is set as $\Delta z = 40$ mm. In Figure 8b, $\Delta x_1$ is reduced to 2.5 μm and each EI contains $120 \times 120$ pixels. The hogel size $p$ is increased to 0.6 mm, so that adjacent hogels will have overlap. As a result, Figure 8b shows better reconstruction quality at the CDP (see the foreleg of the bee), but with degraded reconstruction quality of the school badge. It just verifies the above analysis that the increased hogel size will cause larger defocused spots away from the CDP. In Figure 8c, each transformed hogel is first multiplied with an aperture filter to reduce its size to 0.3 mm; then, all the filtered hogels are spliced seamlessly to form the final hologram. It shows the best reconstruction quality at the marginal depth (the school badge), and the reconstruction quality at the CDP is better than that in Figure 8a. A slight degradation at the CDP can be observed compared with Figure 8b. That is because the numerical filtering causes limited aperture diffraction. Since a spherical wave phase is multiplied with each hogel, the diffraction pattern at the CDP can be thought of as the Fraunhofer diffraction of the rectangular hogel. The zero-order diffraction contains the most energy, with width of $2\lambda L/p$. The zero-order diffraction width determines the upper limit of the resolution. The filtering process reduces the hogel

size $p$, resulting in a degraded diffraction spot. To further improve the reconstruction quality within the total depth range, a partial aperture filtering technique is proposed here. That is, part of the EIs are converted to hogels without aperture filtering to ensure good reconstruction quality near the CDP, and the rest of the EIs are converted to hogels with aperture filtering to improve the reconstruction quality away from the CDP. This classification process can be easily realized in a simple 3D scene such as Figure 5a. The bee model and the school badge are located at the left-half and right-half spaces, respectively. Thus, the EIs mainly contributing to the two objects can be simply classified to the left half and the right half. Then, the left half EIs are transformed to hogels without aperture filtering, while the right half EIs are transformed to hogels with aperture filtering. It is noted that the reconstruction brightness is decreased due to the aperture filtering. Thus, to balance the brightness of the left and right halves, the calculated right half hologram needs to multiply with "2". Through this simple classification, the negative influence of aperture filtering on the reconstruction near the CDP can be relieved, as shown in Figure 8d. Figure 9 shows the optical reconstruction results, which verify the conclusion in Figure 8.

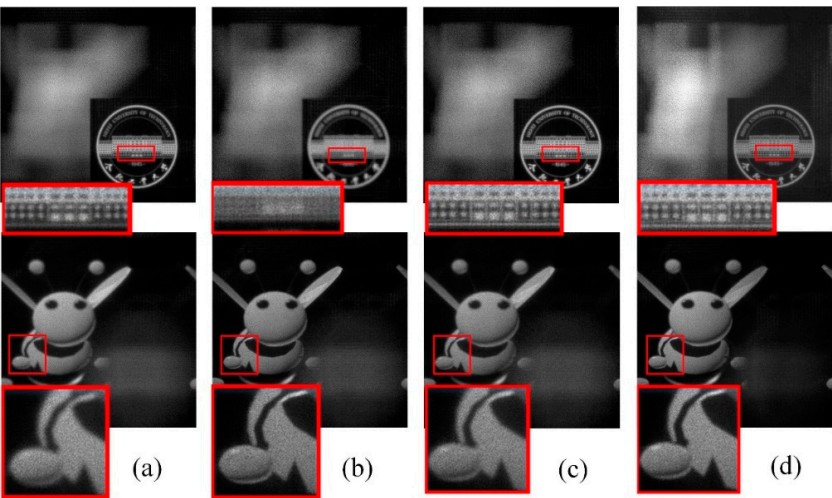

**Figure 8.** (**a**) Numerical simulation results with pixel size $\Delta x_1 = 5$ μm. (**b**) Numerical simulation results with pixel size $\Delta x_1 = 2.5$ μm. (**c**) Numerical simulation results with pixel size $\Delta x_1 = 2.5$ μm and aperture filtering. (**d**) Numerical simulation results with pixel size $\Delta x_1 = 2.5$ μm and partial aperture filtering.

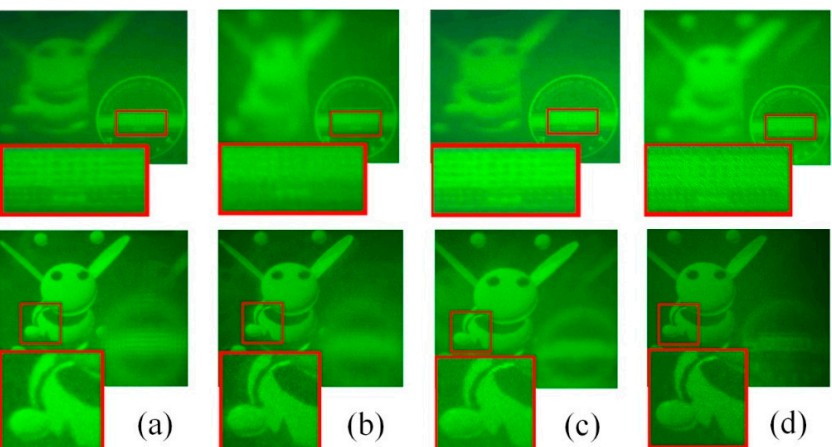

**Figure 9.** (**a**) Optical reconstruction results with pixel size $\Delta x_1 = 5$ μm. (**b**) Optical reconstruction results with pixel size $\Delta x_1 = 2.5$ μm. (**c**) Optical reconstruction results with pixel size $\Delta x_1 = 2.5$ μm and aperture filtering. (**d**) Optical reconstruction results with pixel size $\Delta x_1 = 2.5$ μm and partial aperture filtering.

## 5. Increasing Focal Length *f* with Aperture Filtering and Moving Array Lenslet Technique

Increasing focal length f will decrease the image spot size near the CDP, resulting in better reconstruction quality. However, according to Equation (2), increased *f* will cause increased pixel number *N* of EI. That is, both sizes of hogel and EI increase and cause wider light rays, making larger defocused spots away from the CDP. It can be easily understood by the second term in Equation (5). The above aperture filtering technique can also be used here to narrow the light ray width, as shown in Figure 10b. However, dark gaps among hogels will appear. To fill in the dark gaps, the moving array lenslet technique [23,26] is introduced in Figure 10c. That is, two groups of EIA are captured with lateral shift of $p/2$. Each group of EIA is converted into the corresponding sub-hologram, and the two sub-holograms have relative shift in real space. Then, the two holograms are superimposed with relative shift of $p/2$. The dark gaps of one hologram will be filled up by the other hologram. Note that considering the two-dimensional case, four groups of EIA are needed.

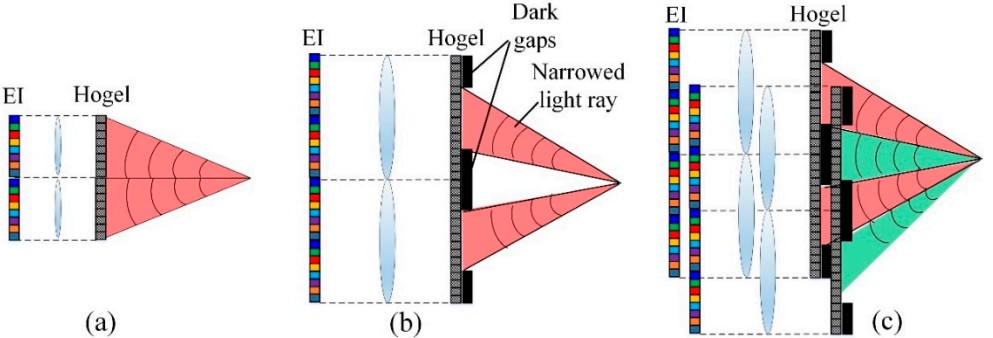

**Figure 10.** (**a**) A conventional case with a small focal length. (**b**) Increased focal length with aperture filtering, resulting in dark gaps. (**c**) Increased focal length with aperture filtering and MALT.

Figure 11a shows the same simulation results as Figure 8a, with focal length of 3 mm. In Figure 11b, the focal length *f* is increased to 6 mm, and the sizes of both hogel and EI increase to 0.6 mm, which means the number of hogels or EIs decreases to $30 \times 30$. Figure 11b shows that increasing *f* improves the reconstruction quality near the CDP, at the price of quality degradation at the marginal depth. The aperture filtering is applied to the right half hogels in Figure 11c, but dark gaps appear. Note that the dark gaps do not appear at the school badge plane. However, when observing the 3D image, our eye will tend to focus on such periodic structures, causing severe image quality degradation. Figure 11d shows the simulation results with MALT, in which the dark gaps disappear. The image quality is improved without decreasing the depth range. Figure 12 shows the optical reconstruction results, which verifies the conclusion in Figure 11.

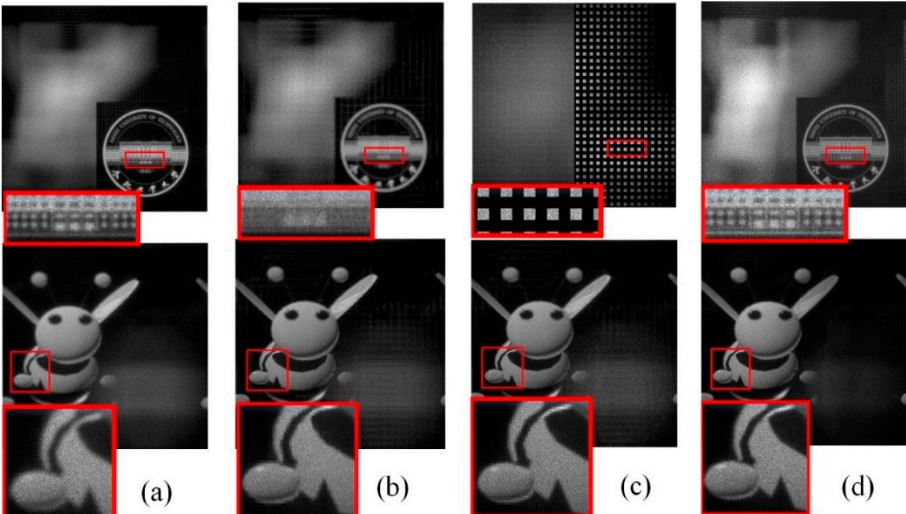

**Figure 11.** (**a**) Numerical simulation results with focal length $f$ = 3 mm. (**b**) Numerical simulation results with focal length $f$ = 6 mm. (**c**) Numerical simulation results with focal length $f$ = 6 mm and partial aperture filtering. (**d**) Numerical simulation results with partial aperture filtering and MALT.

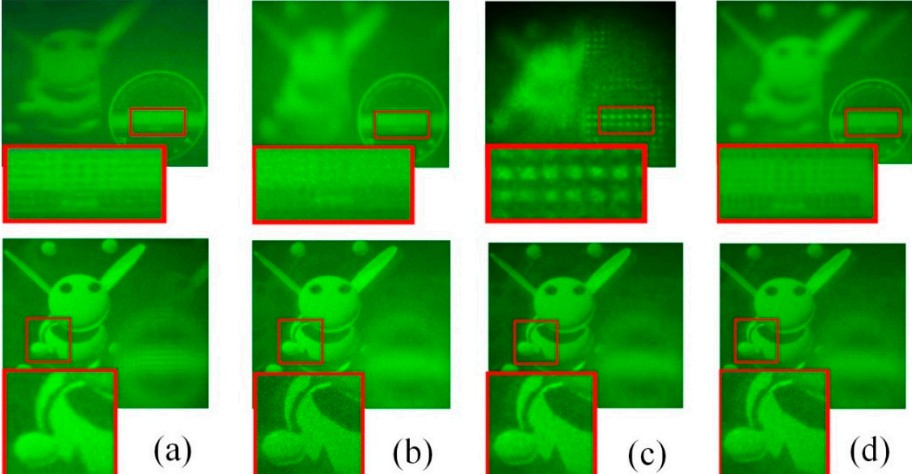

**Figure 12.** (**a**) Optical reconstruction results with focal length $f$ = 3 mm. (**b**) Optical reconstruction results with focal length $f$ = 6 mm. (**c**) Optical reconstruction results with focal length $f$ = 6 mm and partial aperture filtering. (**d**) Optical reconstruction results with partial aperture filtering and MALT.

## 6. Conclusions

In this paper, the resolution of RPHS was analyzed in a geometrical imaging model. Three parameters, capturing depth, pixel size of EI and focal length of the lens array, mainly determine the resolution. Firstly, reducing capturing depth is always beneficial to the resolution enhancement near the CDP. To avoid decreased depth range, a proper capturing depth needs to be chosen through a numerical method. Secondly, reducing pixel size of EI is always effective for the resolution enhancement near the CDP. However, the increased hogel size will cause wider light rays, making larger defocused spots away from the CDP. A partial aperture filtering technique is proposed to enhance resolution within the total depth range. Lastly, increasing focal length of the lens array will always improve the reconstruction quality near the CDP. However, the sizes of EI and hogel also increase, resulting in a decreased depth range. The partial aperture filtering can be used to limit the light ray width, but dark gaps appear. The moving array lenslet technique is introduced to fill in the dark gaps. Compared with integral imaging 3D display, the advantage of free wavefront control is shown in the proposed method. In

the integral imaging display, each display unit is identical and hard to control separately. The trade-off between the resolution, depth range and view angle in integral imaging can only be improved by spatial-multiplexing and time-multiplexing techniques. This trade-off can be easily improved through multiplexing-encoding hologram. Simulation and optical experiments were performed to verify the proposed methods. This paper provides resolution enhancement for RPHS from three different aspects. It is a useful solution for quality improvement in different capturing conditions.

**Author Contributions:** Conceptualization, Z.W. and M.X.; writing—original draft preparation, Z.W.; writing—review and editing, Z.W., A.W. and H.M.; supervision, G.L.; project administration, Q.F. All authors have read and agreed to the published version of the manuscript.

**Funding:** This research was funded by National Natural Science Foundation of China, grant number 61805065 and 61805066; The Fundamental Research Funds for the Central Universities, grant number JZ2021HGTB0077.

**Conflicts of Interest:** The authors declare no conflict of interest.

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
