# Peer review of "Resolution Enhancement of Spherical Wave-Based Holographic Stereogram with Large Depth Range†"

_applsci, doi:10.3390/app11125595_

Round 1

Reviewer 1 Report

n this paper, we propose a method to improve the image quality of the reconstructed image in the central depth plane by using virtual spherical waves. We also derive a geometrical optics method to extend the depth range of the reconstructed image. The proposed method is interesting and the paper is well-written. This paper can be fully adopted if the following minor comments are addressed. reconstructed
1. The key equation in this work is Equation 4. This equation is derived by using geometrical optics based on Figure 2, but it may be difficult for the reader to derive this equation from Figure 2. Therefore, the derivation of Equation 4 should be done more carefully.
2. The parameter "S" should be written in Figure 2 so that the reader can see where "S" is in Figure 2. 3.
3. In lines 207 and 214 on page 7, the variable p is defined as different meanings. One of the variables must be changed.
4. In this study, numerical reconstruction is required to find the appropriate capturing depth. This will be a major obstacle for the real-time playback of this method. In addition, how much time is needed for this numerical reconstruction?

Author Response

In this paper, we propose a method to improve the image quality of the reconstructed image in the central depth plane by using virtual spherical waves. We also derive a geometrical optics method to extend the depth range of the reconstructed image. The proposed method is interesting and the paper is well-written. This paper can be fully adopted if the following minor comments are addressed. reconstructed
1. The key equation in this work is Equation 4. This equation is derived by using geometrical optics based on Figure 2, but it may be difficult for the reader to derive this equation from Figure 2. Therefore, the derivation of Equation 4 should be done more carefully.

Response: We thank the reviewer for his supportive and constructive comments. We added the following sentences to explain the derivation of Eq. (4) more clearly.

The image spot size S at reconstruction depth z can be derived easily by dividing the defocused spot to two parts: the green part and the red part. The green part is related with the focused image spot (represented by the blue rectangle) in a similar triangle relation, and the red part is related with the hogel size p in another similar triangle relation.

  1. The parameter "S" should be written in Figure 2 so that the reader can see where "S" is in Figure 2. 3.

Response: The parameter "S" is added in Figure 2.

  1. In lines 207 and 214 on page 7, the variable p is defined as different meanings. One of the variables must be changed.

Response: The second p is changed to p0. The revision is marked in red in the following sentence.

The calculated complex amplitude distribution of each hogel is multiplied with a rectangle function Rect(u/p0)× Rect(v/p0), where p0 is the pitch of the EI.

  1. In this study, numerical reconstruction is required to find the appropriate capturing depth. This will be a major obstacle for the real-time playback of this method. In addition, how much time is needed for this numerical reconstruction?

Response: The appropriate capturing depth is easily found by studying the relationship between L and image spot size, as shown in Fig. 3(b). Numerical reconstruction is not needed, because the relationship is directly obtained through Eq. (4). The time needed can be ignored.

Reviewer 2 Report

An ideal 3D display is achievable if the light emerging from objects can be completely reproduced as the objects are actually present. In the early years of imaging it was shown that this can be achieved by so-called integral photography which is obtained by recording multiple images with the help of small-lens array. Transferring this principle to holography the Holographic Stereograms (HS) are produced consisting of array of elemental holograms – hogels.

This paper deals with enhancing the resolution of image reconstruction of HS without decreasing the depth range. The calculations follow their previously introduced “resolution priority HS”, where each transformed hogel is multiplied by converged spherical wave phase. The authors notice that resolution (given by Eq. 5) of each hogel reconstruction depends on the distance of reconstruction plane L, focal length f, and pixel size Δx. In the main part of the paper, those parameters are varied, and the results of numerical simulation and optical reconstruction are presented.

Some optimal parameters are given and discussed in the Conclusion section.

The paper is well written and easy to understand. A fair amount of literature in the Introduction section. The methods applied to obtain results are well described. I would like to see some stronger conclusions in the last section. I did not find any apparent mistakes in the manuscript.

Author Response

An ideal 3D display is achievable if the light emerging from objects can be completely reproduced as the objects are actually present. In the early years of imaging it was shown that this can be achieved by so-called integral photography which is obtained by recording multiple images with the help of small-lens array. Transferring this principle to holography the Holographic Stereograms (HS) are produced consisting of array of elemental holograms – hogels.

This paper deals with enhancing the resolution of image reconstruction of HS without decreasing the depth range. The calculations follow their previously introduced “resolution priority HS”, where each transformed hogel is multiplied by converged spherical wave phase. The authors notice that resolution (given by Eq. 5) of each hogel reconstruction depends on the distance of reconstruction plane L, focal length f, and pixel size Δx. In the main part of the paper, those parameters are varied, and the results of numerical simulation and optical reconstruction are presented.

Some optimal parameters are given and discussed in the Conclusion section.

The paper is well written and easy to understand. A fair amount of literature in the Introduction section. The methods applied to obtain results are well described. I would like to see some stronger conclusions in the last section. I did not find any apparent mistakes in the manuscript.

Response: We thank the reviewer for his supportive and constructive comments. The following sentences are added in the conclusion part.

Compared with integral imaging 3D display, the advantage of free wavefront control is shown in the proposed method. In the integral imaging display, each display unit is identical and hard to control separately. The trade-off between the resolution, depth range and view angle in integral imaging can only be improved by spatial-multiplexing and time-multiplexing techniques. This trade-off can be easily improved through multiplexing-encoding hologram.
